# Exploring the experiences of cancer patients: What drives them to seek treatment outside their residential area and what are the experiences resulting from that decision? A qualitative study

**Jeehee Pyo[1,2,3], Mina Lee[1], Haneul Lee[4], Minsu Ock[1,4,5]***

**1** Task Forces to Support Public Health and Medical Services in Ulsan Metropolitan City, Ulsan, Republic of Korea, **2** Department of Preventive Medicine, Asan Medical Institute of Convergence Science and Technology, Asan Medical Center, University of Ulsan College of Medicine, Seoul, Republic of Korea, **3** Always be with you (The PLOCC Affiliated Counseling Training Center), Seoul, Republic of Korea, **4** Department of Preventive Medicine, Ulsan University Hospital, University of Ulsan College of Medicine, Ulsan, Republic of Korea, **5** Department of Preventive Medicine, University of Ulsan College of Medicine, Seoul, Republic of Korea

* ohohoms@naver.com

## Abstract

### Background

The centralizing cancer care has been a persistent trend, often justified by the volume-outcome relationship. However, this trend raises concerns about potential negative impacts, such as increased patient travel burden, treatment delays, and worsened regional disparities in cancer care. Consequently, there is a growing need for the establishment of a regional comprehensive cancer care system to minimize these disparities. In this study, we explored the treatment experiences of cancer patients who received care at medical institutions outside their residential areas to understand their overall experiences with cancer care and identify areas for improvement in the healthcare system.

### Methods

The participants in this study were 7 residents of Ulsan Metropolitan City who had experienced hospitalization for cancer treatment at a medical institution in another region. In-depth interviews were conducted with each participant for about an hour, exploring the participants' experiences in the process of cancer diagnosis, treatment, and follow-up management. A semi-structured guide was used for in-depth interviews.

### Results

The participants experienced fear after receiving an unexpected possibility of cancer diagnosis as a biopsy result. They wanted a definitive diagnosis as soon as possible, which was not realistic, as a tertiary general hospital in Ulsan featured waiting times of at least 6

**Data availability statement:** All relevant data are within the manuscript and are also available from this URL (repository name: Out-of-area cancer treatment experiences): https://osf.io/jhd5g/.

**Funding:** This work was supported by the Task Forces to Support Public Health and Medical Services in Ulsan Metropolitan City (no grant number). The funders had no role in study design, data collection and analysis, decision to publish, or preparation of the manuscript.

**Competing interests:** The authors have declared that no competing interests exist.

months. Participants were overwhelmed with anxiety, and continued searching for information on the disease by themselves. Most of the processes of cancer diagnosis, treatment, and follow-up management at medical institutions in other regions were a series of hardships. Participants had partially recovered, but were still concerned about becoming unwell. Participants stated that reliable medical institutions in the region and sufficient information related to cancer are needed to improve the quality of life of cancer patients.

## Conclusion

The results of this study reveal that cancer patients face various challenges throughout their long journey of treatment. To establish a comprehensive regional cancer care system, it is necessary to expand the availability of quality cancer care across all regions, strengthen the coordination function of primary care institutions, and develop post-discharge cancer management systems using patient-reported outcomes.

## Introduction

Cancer is a principal cause of burden of disease across the world, with one in six ultimately dying to it [1–3]. In response, countries alongside the World Health Organization, have enacted or endorsed various policies and strategies aimed at mitigating this burden of disease [4–6]. The development of infrastructure that enables timely diagnosis and treatment of cancer is pivotal in lowering the mortality rate of treatable cancers by ensuring that patients receive optimal care [7]. Similarly, nations such as South Korea and Japan have been proactively working to establish the necessary infrastructure for timely diagnosis and optimal treatment of cancer [8,9].

However, the need for complex and highly specialized outpatient and inpatient facilities for cancer diagnosis and treatment has led to a trend towards centralizing cancer care, aiming to provide patients with specialized services [10,11]. This trend has been justified by the "volume-outcome relationship," which posits that a higher caseload results in better treatment outcomes [12,13]. However, centralization has sparked concerns regarding the increased burden on patients from travel, the exacerbation of health inequities, challenges in accessing timely care, decreased treatment adherence, and, ultimately, a decline in the overall quality of cancer care [7,14,15]. Consequently, devising an optimal cancer service system that leverages the advantages of centralization while addressing regional disparities in cancer care has emerged as a pivotal policy challenge [16–18]. This underscores the need for the establishment of self-contained regional cancer care systems, each meticulously tailored to meet the unique needs of its respective region [8].

To establish such a self-contained system for each region, it is essential to analyze healthcare utilization patterns, travel distances, and waiting times for cancer patients by type and region [19,20]. In South Korea, the affinity of cancer types to regions is continuously monitored [21,22], and the resultant data serve as evidence for establishing a regionally self-contained cancer care system. However, recent findings on relevance index for cancers indicate a significant centralization of cancer care in South Korea [22], with a particularly notable concentration of cancer patients in certain major hospitals in Seoul [23]. Therefore, understanding the reasons why patients seek cancer care from healthcare providers outside their residential area, as well as their experiences during this process, is crucial for designing a regionally self-contained cancer care system in South Korea that features accurate and sophisticated policies [24].

Many studies, primarily conducted in Western countries, have employed qualitative research methodologies to examine experiences of traveling for cancer care [25]. However, qualitative investigations of why patients seek treatment at hospitals outside their residential areas that focus East Asian countries, including South Korea, are lacking. Hence, we explored the treatment experiences of cancer patients who received care at medical institutions outside their residential areas to understand their overall experiences with cancer care and identify areas for improvement in the healthcare system.

## Materials and methods

### Study design

This study conducted a qualitative research with 7 participants, and adhered to the standard reporting guidelines of the COREQ (Consolidated Criteria for Reporting Qualitative Research) checklist [26].

### Research team composition

The research team comprised four multidisciplinary members: a male preventive medicine specialist, a female doctoral candidate in medical science, a female nurse, and a female general counseling Master's graduate. Each researcher has extensive experience in conducting qualitative research using content analysis, qualitative case studies, photo-voice methodologies, and consensual qualitative research. All four researchers have published qualitative research in SCI (Science Citation Index) journals.

### Participant selection

The inclusion criteria for the study were as follows: patients (i) aged 19 years and older, (ii) residing in Ulsan Metropolitan City, (iii) with experience of hospitalization for cancer treatment in a medical institution outside their residential area (based on city), and (iv) capable of unimpaired communication. Ulsan is administratively divided into four districts (Gu) and one county (Gun), with a total population of approximately 1.1 million as of July 2024. The city has one tertiary hospital, eight general hospitals, and one public healthcare facility. The regional cancer center is located within the tertiary hospital. South Korea is composed of 17 administrative regions, and Ulsan is one of them. These selection criteria refer to individuals who accessed medical institutions in the other 16 regions outside of Ulsan for the purpose of cancer treatment.

A convenience purposive sampling method was utilized to select participants aligned with the research goals [24]. Specifically, the researchers posted a recruitment notice on the Ul-fam (울팸, Ulsan Public Healthcare Family) website, managed by the Task Forces to Support Public Health and Medical Services in Ulsan Metropolitan City [27]. Participant recruitment was conducted from August 31, 2022 to September 9, 2022. Sixteen individuals expressed their interest through the website. Subsequently, the research team selected 11 participants based on additional selection criteria such as cancer type and stage, hospitalization at medical institutions outside the region, sex, and age. After direct communication with these individuals to discuss the study's details, 7 who agreed to participate and were able to arrange their interview schedules were selected as participants (Fig 1).

### Data collection methods and procedures

In this study, a content analysis methodology that incorporates both qualitative and quantitative approaches was employed to derive valid inferences from the collected data [28,29]. This

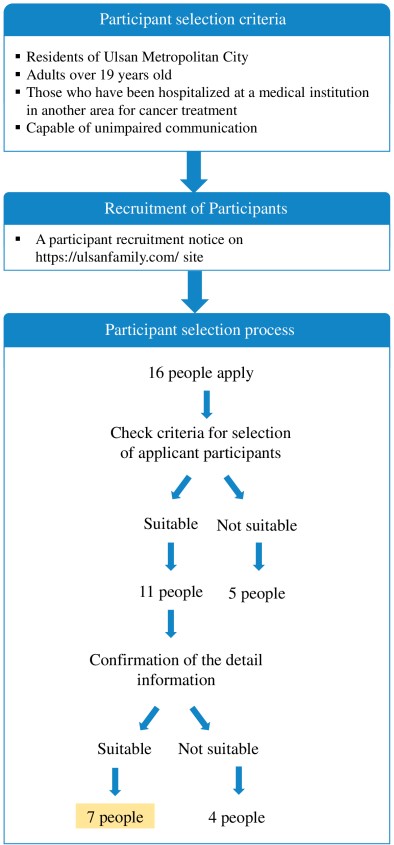

**Fig 1. Participant selection process.**

study utilized the conventional content analysis approach an inductive approach that derives categories directly from the collected data [29]. Researchers gathered data using telephone and in-depth interviews with participants. Adhering to ethical considerations, the researchers fully explained the study's goals and procedure to the participants, obtaining their written consent for participation and screen recording (using online platforms like ZOOM for remote interviews) or audio recordings in-depth interviews (for in-person interviews). For remote interviews, participants were encouraged to choose a location where they felt comfortable sharing their experiences. The researchers conducted remote interviews at a private space within their office. In-person interviews were conducted at separated locations to ensure participants felt comfortable. Participants' experiences were initially gathered using brief telephone interviews, followed by more comprehensive in-depth interviews. Telephone interviews lasted approximately 15 minutes, while in-depth interviews lasted approximately 1 hour.

In-depth interviews were conducted based on a semi-structured guideline, focusing on participants' experiences with medical institutions located outside of Ulsan. A preventive medicine specialist from the research team assessed the suitability of the interview questions. The interview protocol included specific questions such as: 1) "What experiences did you have during the cancer diagnosis process?", 2) "What experiences did you have during the cancer treatment process?", and 3) "What are your current experiences of cancer management?" These questions aimed to elicit comprehensive insights into the psychological, physical, and social experiences of the participants, particularly as cancer patients who sought treatment at

medical institutions in other regions. Data were collected between September 28 and October 12, 2022.

## Analysis methods and procedures

The transcripts from the in-depth interview recordings served as the data source for analysis. Specifically, the 3 researchers who conducted the in-depth interviews meticulously read all the transcripts, employing a line-by-line approach to extract "meaning units.". Each researcher independently identified "meaning units," which were validated through consensus among the team. Subsequently, one researcher organized these meaning units into major themes (primary categorization), labeled these categories to reflect the shared experiences among participants within these themes (secondary categorization), and arranged each category into subcategories based on temporal sequence and relevance, thus providing overarching category names (tertiary categorization). The results of this three-tiered categorization were reviewed by the other 2 researchers involved in the in-depth interviews, ensuring accuracy and consensus.

## Data validation

Data validity was rigorously tested against Guba and Lincoln's established criteria: truth value, applicability, consistency, and neutrality [30]. First, the inquiry's truth value was validated by ensuring that the category outcomes accurately reflected the experiences of all 7 participants. Second, the applicability of the findings was secured by continuing data collection until no new insights emerged and theoretical saturation was achieved during analysis. Third, to ensure consistency, the research process was meticulously documented, with the team actively seeking guidance from seasoned qualitative research experts. Finally, to maintain neutrality, researchers rigorously documented and reviewed their initial understandings and potential biases regarding the research topic, employing bracketing techniques throughout the study.

## Ethical considerations

This study received ethical approval from the Institutional Review Board (IRB) of Ulsan University Hospital (IRB No.: 2021-05-042). All participants provided written informed consent to participate.

## Results

Detailed information regarding the sociodemographic and cancer-related characteristics of the participants is provided in Table 1.

Table 1. Sociodemographic and cancer-related characteristics of the participants.

| No. | Sex | Age | Residential district | Target region | Cancer type | Stage | Number of general hospital-level medical institutions within the residence | Time required to reach local cancer center by public transportation |
|---|---|---|---|---|---|---|---|---|
| 1 | F | 30s | Dong-gu | Seoul | Breast cancer | 2 | 1 | Less than 30 minutes |
| 2 | F | 40s | Buk-gu | Seoul | Thyroid cancer | 2 | 2 | 45 minutes |
| 3 | F | 50s | Ulju-gun | Seoul | Thyroid cancer | 1 | 2 | Over 1hr |
| 4 | F | 40s | Ulju-gun | Seoul | Breast cancer | 1 | 2 | |
| 5 | F | 50s | Ulju-gun | Busan | Thyroid cancer | 3 | 2 | |
| 6 | F | 30s | Ulju-gun | Seoul | Parotid gland cancer | 2 | 2 | |
| 7 | F | 40s | Buk-gu | Seoul | Breast cancer | 1 | 2 | 45 minutes |

The analysis of the transcribed data from each participant resulted in a total of 416 meaning units which were ultimately categorized into 9 subcategories, grouped into 3 main categories (Table 2).

## Seeking cancer treatment beyond local options

The participants found the process of choosing a hospital amid the uncertainty of a cancer diagnosis to be quite complex. The experience of facing long waits at local hospitals caused anxiety and frustration. A lack of trust in local medical institutions acted as a catalyst, prompting them to search for the best possible treatment options. Furthermore, recommendations from friends and acquaintances also motivated participants to look for medical institutions reputed for expertise specific to treating their illness.

**Contrary to my desperate feelings, the recommended hospital in my residence area required a long wait.** Most participants were confronted with unexpected malignancy suspicions during screenings. During this stressful situation, being referred to a local major hospital resulted in slim hope and deepening anxiety. Participants described an urgent desire to verify the malignancy of the lesion and initiate treatment promptly if necessary. However, accessing treatment at local major hospitals proved difficult. Contrary to the participants' desperate hopes, local major hospitals often featured waiting times of several months for treatment or simply registering. For those facing a potential cancer diagnosis, this unforeseen obstacle of prolonged waiting periods presented an insurmountable challenge.

*"When they say, 'Look somewhere else,' well,... it's frustrating. It's bewildering. Usually, you call right after getting diagnosed with cancer. You don't wait for a couple of days to call after some rest. That day, you're tense... There was no need to be that dismissive. They could have asked, 'Would you like to wait?' But it was discouraging to hear that it's not possible at all."* (Participant 2)

*"I thought I might get an appointment in about a month, but when I heard it would take about two months... That seemed too long."* (Participant 6)

Table 2. Categorization of participants' experiences.

| Main categories | Subcategories |
|---|---|
| 1. Seeking cancer treatment beyond local options | 1-1. Contrary to my desperate feelings, the recommended hospital in my residence area required a long wait |
| | 1-2. Unable to entrust cancer treatment due to questioning hospitals' expertise in my residence area |
| | 1-3. Exploring specialized hospitals outside of residence area following frequent advice from friends and family |
| 2. Despite sacrifice and hardship, recovering through cancer treatment at a hospital outside of residence area | 2-1. Anxiety eased by quick treatment scheduling |
| | 2-2. Challenges of traveling for long-distance treatment |
| | 2-3. Family support as a cornerstone of recovery |
| | 2-4. The role of patient-centered care in recovery |
| 3. Difficulty finding suitable medical facilities in residence area for ongoing health management | 3-1. Settling for passive care due to lack of a local primary physician |
| | 3-2. Discomfort at a hospital in residence area discomfort prompting a search for alternatives |

**Unable to entrust cancer treatment due to questioning hospitals' expertise in my residence area.** In regions with a scarcity of local major hospitals, participants found themselves with limited options, primarily due to accumulated distrust in the expertise of local hospitals from past experiences. One participant, dealing with a child's illness, had long perceived a deficiency in the system and expertise at local hospitals. Another participant was influenced by her husband, a lifelong resident of this area, who firmly believed local hospitals were not suitable for treating her illness. Additionally, encounters with unfriendly medical staff further dissuaded participants from seeking care for critical conditions like cancer at local healthcare providers. Despite not having personally experienced negative incidents at local hospitals, one participant sought a second diagnostic opinion from a major hospital outside their local area for added assurance. Ultimately, the low level of trust in local medical institutions was the reason why participants sought care at institutions outside their residential area.

*"My child received leukemia treatment at Hospital A (a major hospital in another region)... Just before that, we had a brief stay at Hospital B (a local major hospital). That's when I really noticed differences in the hospital systems."* (Participant 7)

*"Honestly, I didn't even consider the hospital in my residential area... I went there after a referral from a local hospital, but they actually misdiagnosed me. I didn't know it at the time..."* (Participant 1)

*"I carry the BRCA gene mutation and reached out to a local hospital for information, but they seemed to be unfamiliar with it. This made me think that there might be a significant difference in knowledge compared to what is available in Seoul..."* (Participant 6)

**Exploring specialized hospitals outside of residence area following frequent advice from friends and family.** The participants did not receive treatment at local hospitals, either voluntarily or otherwise. Consequently, they began to look around and listen to the opinions of those close to them. One participant discovered that most people around her with cancer were traveling to Seoul for treatment. Other participants were told that there was a significant difference in medical technology between the metropolitan area and the provinces. Consequently, participants started searching for medical institutions that specialized in their particular illnesses, choosing their healthcare providers based on factors such as the number of surgeries performed, reputation, the amount of information provided about the disease, and achievements in related research. Specifically, one participant even researched the professors conducting clinical trials on cancer. As a result, the experiences and opinions of those around them, as well as the various achievements demonstrated by medical institutions, were factors influencing participants' choice of medical institutions.

*"Every third person around me goes to Hospital S in another region, while others go to Hospital A, also a non-local hospital. … Around that time, I came across a list of renowned doctors on my phone. I have a thyroid issue, and that hospital is specialized in it. That's how I chose my hospital. It seems that the internet and TV do provide a lot of information to people."* (Participant 4)

*"My brother-in-law told me that medical technology in the provinces is 20 years behind Seoul... Only one professor in the Daegu area was chosen [for clinical trials and projects for small and medium-sized enterprises]. All the rest were in Seoul... It seems that professors in the provinces don't work as hard."* (Participant 1)

## Despite sacrifice and hardship, recovering through cancer treatment at a hospital outside of residence area

The participants managed to quickly schedule their medical appointments at hospitals outside their local area, gaining hope for recovery. However, hospitals in the metropolitan area caused temporal, physical, and financial challenges for most participants due to the significant distance from their residences. They credited their ability to navigate these challenges to the sacrifices their families made. Amidst these sacrifices and hardships, participants were receiving patient-centered care, moving towards recovery.

**Anxiety eased by quick treatment scheduling.** The participants who sought care at reputable medical institutions were able to quickly schedule their appointments, which alleviated their anxiety. One participant who was unable to secure a spot at a local hospital felt immense relief upon receiving prompt treatment after relocating to another region. Another participant highlighted the special scheduling considerations they received as an out-of-region patient. Those who experienced rapid progression to surgery reported satisfaction with their decision, with one participant humorously comparing the swift process to a factory's production line, feeling lucky for the expedited care.

> *"Immediately upon arrival, I underwent a biopsy. The doctor then suggested that we could proceed with the surgery right away. With the surgery rescheduled a month earlier than before, I went straight in for surgery."* (Participant 3)

> *"I was diagnosed and underwent surgery about a month later, I believe. It all happened so quickly that I hardly had time to think."* (Participant 7)

**Challenges of traveling for long-distance treatment.** The journey to receive cancer treatment at a distant hospital tested the participants' endurance. Traveling to another region for care, especially when ill, imposed significant time constraints and physical limitations. Participants often arrived at the hospital already exhausted, only to face lengthy waiting times of nearly 6 hours, while others stayed for over 12 hours. One participant described these hospital visits as a 'continuation of waiting.'

> *"It's far, and I had to wait a month for the appointment, but there were too many people waiting...."* (Participant 5)

> *"I constantly feel inconvenienced because I have to take a day's leave just to go to Haeundae."* (Participant 2)

> *"On the day of chemotherapy, I go for a blood test early in the morning. If it's okay, I get in. But there are many cancer patients—I almost wait for about 6 hours? If I arrive in the morning, I enter chemotherapy around 3 or 4 PM."* (Participant 3)

Cost was another significant challenge for participants. Traveling to and from hospitals in distant regions incurred substantial transportation expenses. Treatment days were particularly taxing, often requiring the presence of a caregiver, further increasing the overall burden. Some participants were so physically depleted by the treatment that immediately returning home was not possible. In these instances, they needed to arrange for accommodation or hospital admissions for recovery in the vicinity. One participant reported that their total expenses were 7 million won. Over time, the cumulative burden of hospital visits began to heavily weigh on the participants.

*"The KTX fare alone is a significant sum... Indeed, despite receiving a payment for my cancer diagnosis, the cost of a 3-week stay in a nearby hospital amounted to 7 million won. Specifically, at Hospital A, where I underwent surgery, the expenses exceeded 2 million won."* (Participant 1)

**Family support as a cornerstone of recovery.** The sacrifices of the participants' families provided unwavering support as they navigated the path to recovery. Many participants relied on the company of their spouses for hospital visits in distant regions. The severe side effects of chemotherapy made the prospect of travelling alone unthinkable. During this ordeal, one participant witnessed their spouse's weight loss due to caregiving stress, which caused feelings of guilt. Participants with children had to entrust them to the care of grandparents, while some received nursing assistance from their sisters. For the participants, their families were both a pillar of support through the arduous journey of cancer treatment process and a source of guilt.

*"After the surgery, my sister quit her job to take care of me... She came up and stayed with me in the hospital for about 4 nights and 5 days."* (Participant 3)

*"My husband accompanies me to the hospital in Seoul, driving us there. I'm truly grateful for his presence. He provides immense help and comfort, always standing by my side."* (Participant 5)

**The role of patient-centered care in recovery.** Through their own determination and the sacrifices made by their families, the participants were able to concentrate on recovering their health. Despite the necessity of traveling long distances for treatment, they felt confidence in their decision. Some participants felt anxious due to a lack of accessible information on their condition in their locality. In contrast, they encountered many patients facing similar challenges in hospitals located in other regions, and sharing experiences and information about cancer with these patients comforted them and helped reduce their anxiety. The kindness and attentiveness of the medical staff at these distant hospitals provided a stark, positive contrast to the deficiencies they encountered at local medical facilities. Furthermore, the opportunity to receive health education tailored to their needs empowered them for better self-care.

*"It seems they are nicer, and they are better in terms of calling or sending messages about treatment appointments [compared to local hospitals]."* (Participant 2)

*"There were so many patients like me, so it was comforting to talk and share information during treatment."* (Participant 5)

*"The hospital outside my residential area provided education in a more friendly way... The major hospital in my region, on the other hand, seemed to lack that kind of kindness and attentiveness."* (Participant 3)

## Difficulty finding suitable medical facilities in residence area for ongoing health management

Upon returning to their residential areas after surgery at hospitals outside their locality, most participants continued radiotherapy and ongoing management, anticipating a more comfortable follow-up care experience due to proximity. However, securing follow-up care at local

hospitals was challenging; participants were occasionally experienced passive care or outright denial of treatment. Additionally, travel fatigue was exacerbated as local major hospitals equipped to offer necessary follow-up care and management were usually situated far from the city center. This experience made participants wonder if seeking treatment at the same hospital from start to finish may have been preferable.

**Settling for passive care due to lack of a local primary physician.** Upon returning to their place of residence, the participants encountered indifferent attitudes from medical staff during their cancer follow-up care at local hospitals. They felt disrespected when basic procedures like disinfection or the removal of a single stitch were denied, on the basis that they were not registered patients at those hospitals. Additionally, when undergoing radiotherapy at local hospitals, they were treated as external patients, complicating any attempts to report side effects or discomfort. Faced with these recurring issues, some participants were stressed and started to believe that receiving diagnosis, surgery, follow-up care, and management at the same hospital would have been preferable.

*"After the surgery, I discovered a remaining stitch that was causing pain and clearly needed removal. However, when I reached out to five local hospitals for help, all refused on the ground that I was not their patient. It was a matter of just one stitch. I even reached out to a hospital in a different region… My frustration grew to the point where I considered the drastic option of moving to another area."* (Participant 1)

*"I needed disinfection, so I reached out to the hospital where I was initially diagnosed [the local hospital], but they declined to assist me. I wondered whether their response would have been different, had I been a regular client of theirs... They suggested I consult a cancer rehabilitation center; however, even there, I encountered hesitation for reasons unclear to me. Consequently, my sister or husband ended up performing the disinfection at home, guided by instructional videos."* (Participant 3)

**Discomfort at a hospital in residence area discomfort prompting a search for alternatives.** One barrier to utilizing local major hospitals for some participants was the inconvenient transportation options in their area, impacting their ability to manage their health upon treatment completion. Considering the necessity of regular follow-up visits, the inconvenience posed by transportation difficulties led some participants to consider seeking care at more accessible facilities.

*"I reside in Songjeong-dong, Buk-gu, where the lack of bus services made it difficult to visit the local major hospital. The journey, longer than anticipated, coupled with the exhaustion from radiation therapy, exhausted me..."* (Participant 5)

Some participants struggled to coordinate their treatment schedules without the aid of medical staff, particularly during attempts to secure radiotherapy appointments within specific time frames. This necessitated a search for hospitals in other regions. One participant noted that for older patients, the process of transferring to another facility, while feasible, could still prove burdensome due to the physical demands of treatment. Concerns over long wait times and inferior facilities intensified apprehensions over referrals to local hospitals. Consequently, participants felt hindered in receiving comfortable care, both in their own locality and elsewhere.

*"It was difficult for me to arrange the schedule on my own... Moreover, following the initial round of radiotherapy, I needed another session within a month. Therefore, while in*

*chemotherapy, I have to keep calling to secure an appointment for the next radiotherapy session."* (Participant 3)

*"I took a look at local hospitals and found several options. However, both the system and the facilities were greatly inferior to the hospital in Seoul where I received treatment. Despite the higher costs, the quality of care and the environment were significantly lacking."* (Participant 6).

## Discussion

In this study, we explored the treatment experiences of cancer patients who received care at medical institutions outside their residential areas to understand their overall experiences with cancer care and identify areas for improvement in the healthcare system. The findings revealed that patients undergoing cancer treatment in non-local hospitals encountered a blend of positive and negative experiences, ranging from healthcare provider selection to subsequent health management (Fig 2).

Some studies have explored the experiences associated with traveling for cancer diagnosis and treatment, i.e., the process of moving to other regions for cancer care [25]. However, such studies have mostly been conducted in Western countries such as Canada, Australia, and the UK. Considering the cultural nuances surrounding cancer [31,32], and the varied healthcare systems worldwide, examining culture-specific reasons for seeking cancer treatment in distant regions is crucial. This study, conducted in Korea, an East Asian country, clarifies the experiences of cancer patients in other East Asian countries with similar cultural backgrounds. This study methodologically differs from prior research in Korea by using a qualitative methodology to explore the reasons patients seek treatment outside their local area and the specific challenges they face [24]. This study is significant as it provides foundational data for developing and improving a regionally self-contained cancer treatment system in South Korea.

One reason cancer patients in this study chose medical institutions in other regions was the long waiting times at the only university hospital in their area. Additionally, a lack of trust

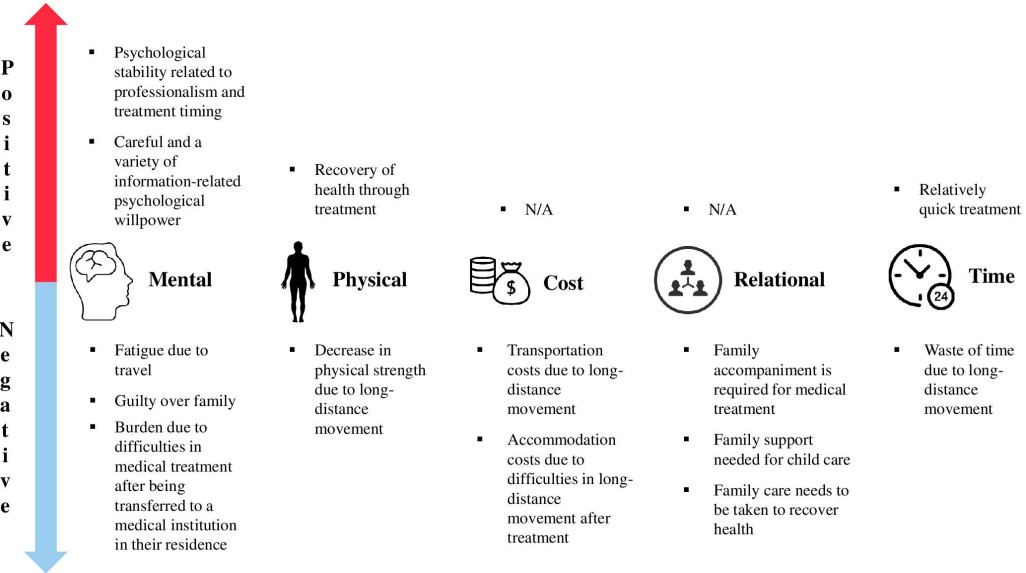

**Fig 2. Positive and negative experiences of patients undergoing cancer treatment at hospitals in regions outside their residential area.**

in local medical institutions also motivated them to seek treatment elsewhere. In this context, this study found that the perception or reality that local hospitals are qualitatively inferior to those in other regions, in terms of timeliness, patient safety, effectiveness, and patient-centeredness, was a major determinant for the choice of non-local medical institutions. This aligns with existing research that suggests that individuals have a tendency to choose safer medical institutions for treatment [33] and that past experiences with medical institutions influence the choice of healthcare provider [34]. Hence, to establish a regionally self-contained cancer treatment system in Korea, the most critical step is expanding the availability of quality cancer care across all regions. This involves a comprehensive approach that includes monitoring and managing waiting times for treatment, incidence rates of adverse events during cancer treatment, survival rates of cancer patients, and patient experience evaluations. These measures are vital for timely, safe, effective, and patient-centered cancer care.

Interestingly, one of the determinants of choosing medical institutions in other regions was the doctors' research achievements and their participation in clinical trials. From the perspective of patients and caregivers, objectively assessing the quality of hospitals is challenging. However, the doctors' research achievements and involvement in clinical trials may act as indicators showcasing the quality level of the respective hospitals. Although Korea implements Quality Assessment Programs for major types of cancer such as colorectal cancer and breast cancer [35,36], insufficient research has been conducted on public awareness, understanding, and the utilization of these programs in choosing medical institutions. This underscores the importance of examining how patients and caregivers perceive the results of the Quality Assessment Program for cancer. Additionally, Korea's Quality Assessment Program for cancer is conducted at the hospital level, rather than the individual physician level. Therefore, public reporting and quality evaluations at the individual physician level are needed for accurate and objective assessments of cancer care physicians by cancer patients.

Similar to previous studies, cancer patients in this study reported various difficulties while seeking treatment in other regions. Being diagnosed with cancer is a devastating situation, and the effort to find an appropriate healthcare provider to receive timely care also posed a significant psychological burden. In Korea, choosing a health provider for post-diagnosis cancer treatment is the patient's decision. The primary care system in Korea is currently underdeveloped, leading to low levels of trust among patients [37,38]. Following a cancer diagnosis, creating a supportive cancer care system capable of guiding patients to the best treatment facilities and offering ongoing management is crucial. This system should also include a comprehensive evaluation of the psychological and social challenges encountered by cancer patients, ensuring they receive appropriate support. Consequently, enhancing the coordinating role of primary care facilities is essential to effectively meet these needs.

Some existing research contends that cancer patients in South Korea receiving treatment at major hospitals are already benefitting from efficient care, leading to potential resistance against the decentralization of cancer care [39]. For such an argument to hold weight, health insurance needs to subsidize transportation expenses that cancer patients incur. However, reaching a social consensus on this issue is complex. As the participants of this study have indicated, cancer patients seeking treatment in different regions must undertake multiple, periodic journeys for chemotherapy, follow-up examinations, and other procedures. These journeys impose financial, physical, psychological, and social difficulties on the patients [40,41]. Additionally, in South Korea, it is a common practice for caregivers to accompany patients on medical visits, which usually leaves cancer patients with profound feelings of guilt [27]. Evaluating the centralization of cancer care through the lens of efficiency alone is insufficient, and there is an urgent need to establish a cancer care system that thoroughly addresses and supports the wide array of challenges cancer patients face.

The cancer patients participating in this study reported satisfaction with both the acquisition of cancer-related information and receiving educational content on managing their condition. In South Korea, the emphasis has traditionally been placed on the outcomes of cancer treatment, often at the expense of thoroughly addressing the unmet needs of patients from their own perspectives. Consequently, there is a need to entrust regional cancer centers with the responsibility of providing precise and essential information spanning from diagnosis to the management of recurrence, which also aids management of their achievements [8]. Additionally, the implementation of a post-discharge cancer management system using patient-reported outcomes is being attempted in Korea, offering a promising avenue to address the unmet needs of cancer patients [42].

This study has limitations concerning the generalizability of its findings, a fundamental problem of qualitative research. Specifically, this study aimed to deeply explore the overall experience of cancer patients receiving care from medical institutions in other regions. While the qualitative research methodology effectively highlighted the common experiences among participants, there are notable limitations regarding the representativeness of the sample. The study sample consists primarily of females aged 30 to 60 with early-stage breast, thyroid, and parotid cancers, excluding other broader cancer populations such as males, individuals aged 60 and above, patients with later-stage or advanced cancer, and those with different types of cancer.

The authors acknowledge that this exclusion limits the generalizability of the findings to the broader cancer population. Future studies are strongly encouraged to employ purposive sampling techniques to include a more diverse range of participants in terms of sociodemographic and clinical characteristics [43]. Such an approach would enhance the representativeness of the study population and provide a more comprehensive understanding of cancer patients' experiences.

## Conclusion

In this study, we conducted a qualitative investigation with cancer patients in Ulsan Metropolitan City who sought cancer care from medical institutions in other regions, exploring their experiences through the entire cancer treatment journey. The key findings highlight challenges such as long waiting times, lack of trust in local hospitals, and the perception of better quality care in other regions. Based on these findings, we recommend policies to improve regional healthcare access, enhance the quality of local cancer care, and establish support systems to assist patients with psychological and social burdens. These recommendations aim to improve the overall healthcare experience for cancer patients in similar contexts.

## Author contributions

**Conceptualization:** Jeehee Pyo, Minsu Ock.

**Data curation:** Jeehee Pyo, Mina Lee, Haneul Lee.

**Formal analysis:** Jeehee Pyo, Mina Lee, Haneul Lee.

**Funding acquisition:** Minsu Ock.

**Investigation:** Jeehee Pyo, Minsu Ock.

**Methodology:** Jeehee Pyo, Minsu Ock.

**Project administration:** Minsu Ock.

**Supervision:** Minsu Ock.

**Validation:** Jeehee Pyo, Mina Lee, Haneul Lee, Minsu Ock.

**Visualization:** Jeehee Pyo, Minsu Ock.

**Writing – original draft:** Jeehee Pyo, Minsu Ock.

**Writing – review & editing:** Jeehee Pyo, Mina Lee, Haneul Lee, Minsu Ock.

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
