## [Decision Letter · Decision Letter 0]

19 Aug 2024

PONE-D-24-15497Experiences of cancer patients who receive cancer treatment outside their area of residence: A qualitative studyPLOS ONE

Dear Dr. Ock,

Thank you for submitting your manuscript to PLOS ONE. After careful consideration, we feel that it has merit but does not fully meet PLOS ONE’s publication criteria as it currently stands. Therefore, we invite you to submit a revised version of the manuscript that addresses the points raised during the review process. Reviewers have highlighted  significant issues to be addressed. Please submit your revised manuscript by Oct 03 2024 11:59PM. If you will need more time than this to complete your revisions, please reply to this message or contact the journal office at plosone@plos.org . Please include the following items when submitting your revised manuscript:

We look forward to receiving your revised manuscript.

Kind regards,

Ali Haider Mohammed

Academic Editor

PLOS ONE

2. We note that your Data Availability Statement is currently as follows: [All relevant data are within the paper and its Supporting Information files.]

Reviewers' comments:

Reviewer's Responses to Questions

**Comments to the Author**

1. Is the manuscript technically sound, and do the data support the conclusions?

Reviewer #1: Yes

Reviewer #2: Partly

2. Has the statistical analysis been performed appropriately and rigorously? 

Reviewer #1: Yes

Reviewer #2: Yes

3. Have the authors made all data underlying the findings in their manuscript fully available?

Reviewer #1: Yes

Reviewer #2: Yes

4. Is the manuscript presented in an intelligible fashion and written in standard English?

Reviewer #1: Yes

Reviewer #2: Yes

5. Review Comments to the Author

Reviewer #1: Thank you for the opportunity to review this manuscript. The authors present an interesting piece of research which I believe holds importance in furthering our understanding as to how place intersects and impact upon cancer care and outcomes. However, I have identified some major and minor concerns that should be addressed prior to being considered for publication.

Major Concerns

It is not clear exactly what this study is trying to achieve. The study's objectives are unclear due to inconsistent articulation of the study aim across the abstract, introduction, and discussion sections. This discrepancy makes it challenging to fully understand the study's intended goals.

The rationale for the study needs to be much better refined. The authors emphasise the need to better understand why patients seek cancer outside their areas of residency. However, as the authors previously highlight in the introduction, the centralization of cancer care in urban areas is clearly a primary factor driving this behaviour. In other words, patients often have no choice but to travel, or as alluded to, there are clear inequalities in healthcare access that perpetuate seeking care elsewhere. It might be more beneficial to rephrase the rationale of the study to reflect the importance of investigating the needs and experiences of cancer patients living in areas where there is limited or poor access to appropriate cancer care.

The study population consists of a sample of female cancer patients aged 30 to 60 with either stage 1 or 2 cancer. This sample is likely not representative of the broader cancer population in the area of Ulsan Metropolitan City. Females in this age range may have unique needs different from those of other genders, age groups, and more advanced cancers. This represents a major concern and significant limitation of the study. Additionally, the recruitment window was open for only 10 days (August 31st – September 9th). Why was this? Extending the recruitment period could have resulted in a more diverse and representative sample.

Abstract

The conclusion of the abstract could be stronger. The study's findings indicate that cancer patients often travel to receive care elsewhere due to various reasons. A more striking and definitive conclusion should emphasise the importance of addressing potential barriers that discourage patients from seeking local treatment.

Introduction

Line 85 ‘succumbing to it’ – what does this mean? Dying because of cancer? This needs to be better clarified.

Methods

The study included a sample size of 7 participants. Whilst it is acknowledged that in qualitative research a sample size of 7 can be considered adequate, it’d like to see more of a rationale for this sample size in context of the research question, the complexity of the problem under investigation, and the method of analysis.

It is crucial to contextualize the setting in which this study is conducted. The core aim is to understand why patients choose to travel elsewhere for cancer care and treatment. Therefore, it is important to provide a comprehensive understanding of the area, including details about the population, hospital infrastructure, and the distance to other cancer centres. This context will greatly assist readers in understanding the study's background and significance.

One of the inclusion criteria was "having experience of hospitalization for cancer treatment in a medical institution outside their residential area." This criterion is somewhat vague; it would be beneficial to clarify what is meant by "outside their residential area." Providing specific details about the distance or geographical boundaries will help readers understand the extent to which patients are traveling for cancer treatment.

Results

The results presented are interesting and offer unique insights into why cancer patients choose to travel outside their area of residence for care. Most findings align with the overarching aim of understanding the reasons behind this travel. However, some results focus more on general experiences and facilitating factors during the patients' journeys. The study's aim should better reflect this broader scope—emphasizing not only why patients travel for care but also their overall experiences and the factors that facilitate their decisions.

Discussion

Overall a good discussion section is presented. My only suggestions for this sections are as follows. 1) the aim needs to be consistent with what is stated in the abstract and introduction, 2) I'd like to see a better acknowledgement of the limitations of the study i.e. the lack of a diverse sample, limited recruitment window etc. and 3) I'd also like to see a brief discussion on how future research can begin to address these issues and start to develop and improve localised cancer services.

Reviewer #2: Dear Authors

I really appreciate your hard work. 

However, significant changes are required to enhance the quality of the current study.

First, the title of the study needs to match the aims of the current study.

The objectives stated in the abstract must match those stated in the introduction.

Furthermore, the current study's introduction was excellent, but it needed to be more focused to make it shorter and more direct to the point.

Methodology

The abstract should be comprehensive and not solely focused on the participants.

Regarding the methodology, we need to provide more specific details about the first two categories.

There is a shortage with the questions used; there must be a category regarding the economic impact, and the causes of travelling to other parts of the country to receive treatment need to be more clearly defined and deepened.

About discussion Please remove the first section; it seems like part of the introduction and methodology.

The same is true about the coincidence.

References

Ok

6. PLOS authors have the option to publish the peer review history of their article (what does this mean? ). If published, this will include your full peer review and any attached files.

**Do you want your identity to be public for this peer review?** For information about this choice, including consent withdrawal, please see our Privacy Policy .

Reviewer #1: No

Reviewer #2: **Yes: ** Bassam Abdul Rasool Hassan

---

## [Author Response · Author response to Decision Letter 0]

25 Sep 2024

Response to reviewers

We would like to express our sincere gratitude for your valuable feedback and constructive comments on our manuscript. We have revised our manuscript in accordance with your suggestions. In the manuscript body, our revisions are in red font to make them easy for you to locate. We look forward to your reply.

[Reviewer 1]

Major Concerns

It is not clear exactly what this study is trying to achieve. The study's objectives are unclear due to inconsistent articulation of the study aim across the abstract, introduction, and discussion sections. This discrepancy makes it challenging to fully understand the study's intended goals.

Response: Thank you for your valuable feedback. This study conducted an in-depth exploration of the experiences of Korean cancer patients who received treatment at medical institutions outside their region. In the introduction of this study, the necessity of the research objective is demonstrated through four main points. First, considering the burden of cancer, it is essential to establish a robust cancer care infrastructure. Second, it is important to develop a regionally tailored cancer care system. Third, there is a need to explore the experiences of cancer patients who seek treatment outside of their residential areas. Lastly, due to the lack of existing research on this topic, this study aims to explore the experiences of cancer patients receiving treatment in different regions using qualitative research methodologies. In the discussion section, we highlighted the need for expanding high-quality cancer care system in each region, the necessity of physician-level quality assessments and public disclosure, and the importance of strengthening the coordination role of primary care in post-cancer management. Based on your feedback, we have reviewed the content and modified the background and discussion in the abstract to better align with the research objectives stated in the introduction and discussion sections of the manuscript (lines 54-61, 76-80).

The rationale for the study needs to be much better refined. The authors emphasise the need to better understand why patients seek cancer outside their areas of residency. However, as the authors previously highlight in the introduction, the centralization of cancer care in urban areas is clearly a primary factor driving this behaviour. In other words, patients often have no choice but to travel, or as alluded to, there are clear inequalities in healthcare access that perpetuate seeking care elsewhere. It might be more beneficial to rephrase the rationale of the study to reflect the importance of investigating the needs and experiences of cancer patients living in areas where there is limited or poor access to appropriate cancer care.

Response: Thank you for your valuable feedback. We have revised the background section in the abstract, and also in the main text, we have emphasized the importance of policies aimed at narrowing the socioeconomic disparities that can clearly arise from the centralization of cancer care by adding further references to support this need (lines 102, reference 16-19).

The study population consists of a sample of female cancer patients aged 30 to 60 with either stage 1 or 2 cancer. This sample is likely not representative of the broader cancer population in the area of Ulsan Metropolitan City. Females in this age range may have unique needs different from those of other genders, age groups, and more advanced cancers. This represents a major concern and significant limitation of the study. Additionally, the recruitment window was open for only 10 days (August 31st – September 9th). Why was this? Extending the recruitment period could have resulted in a more diverse and representative sample.

Response: Thank you for your valuable feedback. I understand your concerns, and I appreciate your thoughtful input. Qualitative research is not intended to provide representativeness but rather to capture and present the experiences of participants as they are. As you mentioned, some participants, particularly those in their 30s, faced unique challenges related to their life stage (e.g., raising children). While the categorization highlights the main common experiences of participants, specific experiences that were particularly significant to certain individuals were written separately. Nonetheless, to avoid any overinterpretation of the findings, I ensured that the study’s limitations are clearly stated (lines 516-520). Regarding the recruitment period, I kindly ask for your understanding that the recruitment was conducted over 10 days due to the overall research timeline.

Abstract

The conclusion of the abstract could be stronger. The study's findings indicate that cancer patients often travel to receive care elsewhere due to various reasons. A more striking and definitive conclusion should emphasise the importance of addressing potential barriers that discourage patients from seeking local treatment.

Response: Incorporating your feedback, we have revised the conclusion of the abstract to emphasize that overcoming various obstacles is essential for establishing a comprehensive regional cancer care system (lines 76-80).

Introduction

Line 85 ‘succumbing to it’ – what does this mean? Dying because of cancer? This needs to be better clarified.

Response: Thank you for your detailed feedback. ‘Succumbing’ has been modified to ‘dying’ (lines 85-86).

Methods

The study included a sample size of 7 participants. Whilst it is acknowledged that in qualitative research a sample size of 7 can be considered adequate, it’d like to see more of a rationale for this sample size in context of the research question, the complexity of the problem under investigation, and the method of analysis.

Response: We made efforts to ensure a transparent recruitment process for participants. Through the recruitment announcement, a total of 16 individuals expressed their willingness to participate. To ensure maximum variation sampling, the research team reviewed the residence, cancer stage, gender, age, and type of cancer of the 16 applicants and selected 11. Of these 11 selected participants, 4 were excluded due to refusal to consent, poor health status, or non-response, leaving a final total of 7 participants. Additionally, during the data collection process, the research team confirmed theoretical saturation regarding the participants’ experiences, and thus, no further participants were recruited.

It is crucial to contextualize the setting in which this study is conducted. The core aim is to understand why patients choose to travel elsewhere for cancer care and treatment. Therefore, it is important to provide a comprehensive understanding of the area, including details about the population, hospital infrastructure, and the distance to other cancer centres. This context will greatly assist readers in understanding the study's background and significance.

Response: Thank you for your detailed feedback. In addition to the sociodemographic information of the participants, we have included the number of general hospitals in their residential areas and the public transportation time from the center of their residence to the regional cancer center to aid the reader’s understanding (line 210, Table 1). In the method section, we also added a footnote providing details on the healthcare status of Ulsan Metropolitan City. Ulsan is administratively divided into four districts (Gu) and one county (Gun), with a total population of approximately 1.1 million as of July 2024. The city has one tertiary hospital, eight general hospitals, and one public healthcare facility. The regional cancer center is located within the tertiary hospital (line 140, footnote 1).

One of the inclusion criteria was "having experience of hospitalization for cancer treatment in a medical institution outside their residential area." This criterion is somewhat vague; it would be beneficial to clarify what is meant by "outside their residential area." Providing specific details about the distance or geographical boundaries will help readers understand the extent to which patients are traveling for cancer treatment.

Response: Thank you for your considerate feedback. ‘Outside of the residential area’ refers to locations other than the participant’s place of residence (on a city-level basis). A more detailed explanation of the participant selection criteria has been provided (line 141). Additionally, we added the following description of Ulsan in Footnote 2. South Korea is composed of 17 administrative regions, and Ulsan is one of them. These selection criteria refer to individuals who accessed medical institutions in the other 16 regions outside of Ulsan for the purpose of cancer treatment.

Results

The results presented are interesting and offer unique insights into why cancer patients choose to travel outside their area of residence for care. Most findings align with the overarching aim of understanding the reasons behind this travel. However, some results focus more on general experiences and facilitating factors during the patients' journeys. The study's aim should better reflect this broader scope—emphasizing not only why patients travel for care but also their overall experiences and the factors that facilitate their decisions.

Response: Thank you for your valuable feedback. In the process of comprehensively exploring the experiences of cancer patients receiving treatment in different regions, we were able to identify a range of experiences, including general experiences and facilitating factors. Upon review, I noticed that while the research objective presented at the end of the introduction addresses the points you mentioned, the objective described in the discussion section was somewhat lacking. I have since revised the research objective in the discussion section to ensure consistency and clarity (lines 436-441).

Discussion

Overall a good discussion section is presented. My only suggestions for this sections are as follows. 1) the aim needs to be consistent with what is stated in the abstract and introduction, 2) I'd like to see a better acknowledgement of the limitations of the study i.e. the lack of a diverse sample, limited recruitment window etc. and 3) I'd also like to see a brief discussion on how future research can begin to address these issues and start to develop and improve localised cancer services.

Response: Thank you for your kind suggestions to improve this research.

1. The aim in the discussion section have been revised to be consistent with those presented in the abstract and introduction (lines 436-441).

2. The limitations have been described in more detail (lines 516-520).

3. We have already proposed various suggestions in the manuscript to encourage localized cancer care. Specifically, we suggested expanding medical institutions capable of providing high-quality cancer treatment within each region, conducting individual physician-level quality assessments to gauge medical expertise and making the results publicly available, strengthening the coordinating role of primary care institutions in cancer care, and enhancing post-discharge cancer management systems for cancer patients.

[Reviewer 2]

First, the title of the study needs to match the aims of the current study.

Response: Thank you for your kind feedback. I have revised the research title to align with the study’s objective of capturing both the reasons why cancer patients seek care at medical institutions outside their residential area and their overall experiences resulting from that decision (lines 1-3).

The objectives stated in the abstract must match those stated in the introduction.

Response: I have revised the objectives stated in the abstract to align them with those specified in the introduction (lines 54-61).

Furthermore, the current study's introduction was excellent, but it needed to be more focused to make it shorter and more direct to the point.

Response: In the introduction of this study, the necessity of the research objective is demonstrated through four main points. First, considering the burden of cancer, it is essential to establish a robust cancer care infrastructure. Second, it is important to develop a regionally tailored cancer care system. Third, there is a need to explore the experiences of cancer patients who seek treatment outside of their residential areas. Lastly, due to the lack of existing research on this topic, this study aims to explore the experiences of cancer patients receiving treatment in different regions using qualitative research methodologies. We have condensed some sections of the introduction to convey the key points more directly (lines 85-86).

Methodology

The abstract should be comprehensive and not solely focused on the participants.

Response: I have comprehensively revised the background and conclusion sections of the abstract. This revision aims to help readers better understand the context of the study and the policy recommendations derived from the research findings (lines 54-61, 76-80).

Regarding the methodology, we need to provide more specific details about the first two categories. There is a shortage with the questions used; there must be a category regarding the economic impact, and the causes of travelling to other parts of the country to receive treatment need to be more clearly defined and deepened.

Response: Thank you for your kind feedback. This study utilized a semi-structured questionnaire. Although only three questions were officially used (lines 169-177), additional experiences were explored based on the flow of participants’ narratives. We already included information related to the economic impact in subcategory “2-2. Challenges of traveling for long-distance treatment” of the results section. Participants reported economic difficulties to a similar extent as physical and psychological challenges, so a separate category for economic impact was not presented. However, we have made efforts to more clearly define the reasons for traveling to other regions for treatment (lines 255-256, 278-280).

About discussion Please remove the first section; it seems like part of the introduction and methodology. The same is true about the coincidence.

Response: We have revised the first section of the discussion extensively based on your feedback. As a result, only the objectives of the study and the results presented in the figures remain (lines 436-441).

---

## [Decision Letter · Decision Letter 1]

17 Nov 2024

PONE-D-24-15497R1Exploring the experiences of cancer patients: What drives them to seek treatment outside their residential area and what are the experiences resulting from that decision? a qualitative studyPLOS ONE

Dear Dr. Ock,

Thank you for submitting your manuscript to PLOS ONE. After careful consideration, we feel that it has merit but does not fully meet PLOS ONE’s publication criteria as it currently stands. Therefore, we invite you to submit a revised version of the manuscript that addresses the points raised during the review process.

Reviewers have addressed certain critical issues that need to be addressed.

We look forward to receiving your revised manuscript.

Kind regards,

Ali Haider Mohammed

Academic Editor

PLOS ONE

Reviewers' comments:

Reviewer's Responses to Questions

**Comments to the Author**

1. If the authors have adequately addressed your comments raised in a previous round of review and you feel that this manuscript is now acceptable for publication, you may indicate that here to bypass the “Comments to the Author” section, enter your conflict of interest statement in the “Confidential to Editor” section, and submit your "Accept" recommendation.

Reviewer #1: (No Response)

Reviewer #2: All comments have been addressed

2. Is the manuscript technically sound, and do the data support the conclusions?

Reviewer #1: Yes

Reviewer #2: Yes

3. Has the statistical analysis been performed appropriately and rigorously? 

Reviewer #1: Yes

Reviewer #2: Yes

4. Have the authors made all data underlying the findings in their manuscript fully available?

Reviewer #1: Yes

Reviewer #2: No

5. Is the manuscript presented in an intelligible fashion and written in standard English?

Reviewer #1: Yes

Reviewer #2: Yes

6. Review Comments to the Author

Reviewer #1: Thank you for addressing my comments. I am largely happy with the revised paper. I only have two minor amendments to recommend.

1. Please attempt to better align the aims across the Abstract, Introduction, and Discussion. I recognise that you have already attempted this, however, they do not fully align. I'd recommend defining one clear aim and repeating it verbatim. The slight variations make it a confusing.

Abstract aim: In this study, we explored the treatment experiences of cancer patients who used medical institution outside their regions, aiming to identify issues within the local cancer care system and propose policy improvements.

Introduction aim: this study aims to conduct qualitative research involving cancer patients to thoroughly investigate their experiences with cancer care at hospitals in different regions, covering the entire care continuum from diagnosis and treatment to management.

Discussion aim: This study conducted a qualitative investigation on cancer patients in Korea to explore their overall experiences with cancer treatment, as well as their experiences receiving treatment at medical institutions located outside their residential area.

2.I fully understand that qualitative research primarily focuses on capturing experiences rather than aiming for objective representation. However, if your sample population is not representative of the broader cancer population, it would be inappropriate to make generalisations about the wider cancer population.

I would like to see a more thoughtful acknowledgment of this in the limitation section. For example 'The findings of this study should be interpreted with caution when generalising to broader cancer populations. The sample consists of females aged 30 to 60, primarily with early-stage breast, thyroid, and parotid cancers. The authors acknowledge the exclusion of other wider cancer populations as a major limitation e.g. those who are male, are older (aged 60 and above), have later-stage or advanced cancer, and have different cancer types. Future studies are strongly encouraged to employ purposive sampling techniques to ensure the study population is more representative of the broader cancer population.'

I would strongly encourage referring to and citing the following article:

Benoot C, Hannes K, Bilsen J. The use of purposeful sampling in a qualitative evidence synthesis: A worked example on sexual adjustment to a cancer trajectory. BMC Med Res Methodol. 2016 Feb 18;16:21. doi: 10.1186/s12874-016-0114-6. PMID: 26891718; PMCID: PMC4757966.

Reviewer #2: Dear Authors,

I would like to express my appreciation for the considerable effort you have put into making the necessary amendments to your manuscript. Your dedication to improving the quality of your work is commendable.

However, I would like to suggest a few further revisions:

Aims in the Introduction: I recommend revisiting and rewriting the aims of your study as presented in the introduction. The current version could be more closely aligned with the study's title, results, and discussion. The aims you included in the abstract appear to be more consistent and reliable in relation to the overall study. Please consider revising the introduction to reflect a similar approach.

Conclusion: It is important to concisely summarize the main findings of your study in the conclusion. In addition, I suggest incorporating recommendations based on your findings, without expanding on what the results may potentially lead to. The focus should remain on the study's key outcomes and their direct implications.

Thank you for your continued efforts, and I look forward to reviewing the revised version.

Best regards,

7. PLOS authors have the option to publish the peer review history of their article (what does this mean? ). If published, this will include your full peer review and any attached files.

**Do you want your identity to be public for this peer review?** For information about this choice, including consent withdrawal, please see our Privacy Policy .

Reviewer #1: No

Reviewer #2: **Yes: ** Bassam Abdul Rasool Hassan

---

## [Author Response · Author response to Decision Letter 1]

29 Dec 2024

Response to reviewers

We would like to express our sincere gratitude for your valuable feedback and constructive comments on our manuscript. We have revised our manuscript in accordance with your suggestions. In the manuscript body, our revisions are in red font to make them easy for you to locate. We look forward to your reply.

[Reviewer 1]

1. Please attempt to better align the aims across the Abstract, Introduction, and Discussion. I recognise that you have already attempted this, however, they do not fully align. I'd recommend defining one clear aim and repeating it verbatim. The slight variations make it a confusing.

Abstract aim: In this study, we explored the treatment experiences of cancer patients who used medical institution outside their regions, aiming to identify issues within the local cancer care system and propose policy improvements.

Introduction aim: this study aims to conduct qualitative research involving cancer patients to thoroughly investigate their experiences with cancer care at hospitals in different regions, covering the entire care continuum from diagnosis and treatment to management.

Discussion aim: This study conducted a qualitative investigation on cancer patients in Korea to explore their overall experiences with cancer treatment, as well as their experiences receiving treatment at medical institutions located outside their residential area.

Response: Thank you for your valuable feedback. Based on your suggestion, we have revised the aims in the abstract, introduction, and discussion to ensure consistency throughout the manuscript (lines 58-61, 119-122, 436-438).

2. I fully understand that qualitative research primarily focuses on capturing experiences rather than aiming for objective representation. However, if your sample population is not representative of the broader cancer population, it would be inappropriate to make generalisations about the wider cancer population.

I would like to see a more thoughtful acknowledgment of this in the limitation section. For example 'The findings of this study should be interpreted with caution when generalising to broader cancer populations. The sample consists of females aged 30 to 60, primarily with early-stage breast, thyroid, and parotid cancers. The authors acknowledge the exclusion of other wider cancer populations as a major limitation e.g. those who are male, are older (aged 60 and above), have later-stage or advanced cancer, and have different cancer types. Future studies are strongly encouraged to employ purposive sampling techniques to ensure the study population is more representative of the broader cancer population.'

I would strongly encourage referring to and citing the following article:

Benoot C, Hannes K, Bilsen J. The use of purposeful sampling in a qualitative evidence synthesis: A worked example on sexual adjustment to a cancer trajectory. BMC Med Res Methodol. 2016 Feb 18;16:21. doi: 10.1186/s12874-016-0114-6. PMID: 26891718; PMCID: PMC4757966.

Response: Thank you for your valuable feedback. We have revised the description of this study's limitations to provide greater clarity. Additionally, using the recommended reference, we emphasized the need for purposive sampling in future studies to address the limitations identified in this research (lines 515-528).

[Reviewer 2]

Aims in the Introduction: I recommend revisiting and rewriting the aims of your study as presented in the introduction. The current version could be more closely aligned with the study's title, results, and discussion. The aims you included in the abstract appear to be more consistent and reliable in relation to the overall study. Please consider revising the introduction to reflect a similar approach.

Response: Thank you for your valuable feedback. Based on your suggestion, we have revised the aims in the abstract, introduction, and discussion to ensure consistency throughout the manuscript (lines 58-61, 119-122, 436-438).

Conclusion: It is important to concisely summarize the main findings of your study in the conclusion. In addition, I suggest incorporating recommendations based on your findings, without expanding on what the results may potentially lead to. The focus should remain on the study's key outcomes and their direct implications.

Response: Thank you for your valuable feedback. In the conclusion, we have concisely summarized the key findings of the study. Additionally, we have revised the content to focus on the specific recommendations derived from the study's results (lines 533-538).

---

## [Decision Letter · Decision Letter 2]

6 Feb 2025

Exploring the experiences of cancer patients: What drives them to seek treatment outside their residential area and what are the experiences resulting from that decision? a qualitative study

PONE-D-24-15497R2

Dear Dr. Minsu,

We’re pleased to inform you that your manuscript has been judged scientifically suitable for publication and will be formally accepted for publication once it meets all outstanding technical requirements.

Kind regards,

Ali Haider Mohammed

Academic Editor

PLOS ONE

Additional Editor Comments (optional):

Reviewers' comments:

Reviewer's Responses to Questions

**Comments to the Author**

1. If the authors have adequately addressed your comments raised in a previous round of review and you feel that this manuscript is now acceptable for publication, you may indicate that here to bypass the “Comments to the Author” section, enter your conflict of interest statement in the “Confidential to Editor” section, and submit your "Accept" recommendation.

Reviewer #1: All comments have been addressed

Reviewer #2: All comments have been addressed

2. Is the manuscript technically sound, and do the data support the conclusions?

Reviewer #1: Yes

Reviewer #2: Yes

3. Has the statistical analysis been performed appropriately and rigorously? 

Reviewer #1: N/A

Reviewer #2: Yes

4. Have the authors made all data underlying the findings in their manuscript fully available?

Reviewer #1: Yes

Reviewer #2: Yes

5. Is the manuscript presented in an intelligible fashion and written in standard English?

Reviewer #1: Yes

Reviewer #2: Yes

6. Review Comments to the Author

Reviewer #1: Thank you for addressing my comments. I'd like to congratulate the authors for conducting an interesting and meaningful piece of work. Well done.

Reviewer #2: (No Response)

7. PLOS authors have the option to publish the peer review history of their article (what does this mean? ). If published, this will include your full peer review and any attached files.

**Do you want your identity to be public for this peer review?** For information about this choice, including consent withdrawal, please see our Privacy Policy .

Reviewer #1: No

Reviewer #2: **Yes: ** Bassam Abdul Rasool Hassan

---

## [Editor Report · Acceptance letter]

PONE-D-24-15497R2

PLOS ONE

Dear Dr. Ock,

I'm pleased to inform you that your manuscript has been deemed suitable for publication in PLOS ONE. Congratulations! Your manuscript is now being handed over to our production team.

Kind regards,

on behalf of

Dr. Ali Haider Mohammed

Academic Editor

PLOS ONE